# Predictive factors for unfavourable treatment in MDR-TB and XDR-TB patients in Rio de Janeiro State, Brazil, 2000-2016

**Marcela Bhering**[1,2]*, **Raquel Duarte**[3,4], **Afrânio Kritski**[1,2]

**1** School of Medicine, Federal University of Rio de Janeiro, Rio de Janeiro, Brazil, **2** Brazilian Tuberculosis Research Network / REDE TB, Rio de Janeiro, Brazil, **3** EPIUnit, Public Health Institute, University of Porto, Porto, Portugal, **4** Public Health Science and Medical Education Department, School of Medicine, University of Porto, Porto, Portugal

* marbhering@gmail.com

## Abstract

### Setting

The State of Rio de Janeiro stands out as having the second highest incidence and the highest mortality rate due to TB in Brazil. This study aims at identifying the factors associated with the unfavourable treatment of MDR/XDR-TB patients in that State.

### Method

Data on 2269 MDR-TB cases reported in 2000–2016 in Rio de Janeiro State were collected from the Tuberculosis Surveillance System. Bivariate and multivariate logistic regressions were run to estimate the factors associated with unfavourable outcomes (failure, default, and death) and, specifically, default and death.

### Results

The proportion of unfavourable outcomes was 41.9% among MDR-TB and 81.5% among XDR-TB. Having less than 8 years of schooling, and being an Afro-Brazilian, under 40 years old and drug user were associated with unfavourable outcome and default. Bilateral disease, HIV positive, and comorbidities were associated with death. XDR-TB cases had a 4.7-fold higher odds of an unfavourable outcome, with 29.3% of such cases being not treated for multidrug resistance in the past.

### Conclusion

About 30% of XDR-TB cases may have occurred by primary transmission. The high rates of failure and death in this category reflect the limitation of treatment options. This highlights the urgency to incorporate new drugs in the treatment.

**Data Availability Statement:** All relevant data are within the manuscript and its Supporting Information files.

**Funding:** The author(s) received no specific funding for this work.

**Competing interests:** The authors have declared that no competing interests exist.

## Introduction

Resistant multidrug TB (MDR-TB) is defined as TB with resistance to at least rifampin and isoniazid and extensively resistant TB (TB-XDR), such as MDR-TB plus resistance to at least one quinolone, and to one second-line injectable drugs used to treat TB (capreomycin, kanamycin and amikacin) [1]. They are a source of a public-health crises worldwide. The treatment is longer, requires more expensive and toxic drugs than TB with drug-sensitive bacilli [1,2].

In Brazil the incidence of TB decreased from 39 per 100 thousand inhabitants in 2008 to 33.5 in 2017. The mortality rate went from 2.6 per 100 thousand inhabitants in 2007 to 2.2 in 2016 [3].

However, the State of Rio de Janeiro (henceforth RJ) stands out as having the second highest incidence in the country and the highest mortality rate due to TB in the country, which were 63.5 in 2017 and 4.4 per 100 thousand inhabitants in 2016, respectively.[3] Given that this state is one of the country's most developed, it is relevant to investigate the reasons behind its poor performance [4].

That said, this study aims to identify factors associated with the unfavourable treatment of patients with MDR/XDR-TB in RJ, also considering default and death as specific outcomes.

## Materials and methods

### Data and sample

This was an observational retrospective cohort study based on secondary data. The cohort was extracted from the Special Tuberculosis Treatment System (SITETB). SITETB is an electronic information system of Brazil's Health Ministry, used for the compulsory notification and follow-up of all TB cases requiring special treatments by patients unable to use the standard TB regimen (2RHZE/4RH). Demographic and clinical data, drug susceptibility testing (DST) results, adverse events, treatment regimens and outcomes for each patient are recorded. Periodically the data are updated according to clinical progress by professionals from the centers of reference [5].

### Ethics statement

The study protocol was approved by the Research Ethics Committee of the Federal University of Rio de Janeiro (CAAE 10126919.2.0000.5257), which granted permission for use of the identified data for the purposes of the study and waived the need for written informed consent from participants as the study was based on secondary data and involved no more than minimal risk. All patients had an identification number, and to protect patients' confidentiality, only one investigator (MB) had access to both identified and de-identified codes; she prepared the anonymous database that was used in the study.

**Tuberculosis control program, treatment regimens for MDR-TB and laboratory diagnosis in Brazil.** Since the 1960s, Brazil has distributed free antituberculosis drugs. The private sector, when diagnosing TB cases, referrals to a public sector and a private pharmacies, do not offer first- or second-line TB drugs. Medicines are centrally ordered and purchased by the Ministry of Health.

Regarding MDR-TB, since 2000 or in Brazil, it has been developing epidemiological actions, with protocol for treatment, notification and follow-up of drug resistance cases. Brazilian guidelines [6] recommend treatment of MDR-TB with standardized regimens, mainly because DST in many settings of the Brazilian public free-of-charge health system is restricted only to first-line drugs [7].The standardized treatment regimen for MDR-TB is recommended and applied in Brazil and should include four drugs, preferably not previously used, containing: a

fluoroquinolone, an injectable drug, two second-line drugs (ethionamide, terizidone, linezolid or clofazimine) and an oral first line drug (ethambutol or pyrazinamide), if susceptible [6].

Individualized regimens are restricted to patients with additional resistance to first-line drugs, pre-XDR, XDR-TB, and to patients who have had adverse events with standardized regimens. These regimens might include other oral drugs, such as clofazimine, linezolid, imipenem and high-dose isoniazid [8]. During the study period, the main change in the treatment regimen was the 2009 recommendation to use streptomycin as an injectable drug in preference to amikacin [9].

Moreover, about laboratory tests, cultures and DST for first line medicines were performed at the Central Laboratory of RJ (LACEN) and DST for second line drugs were performed at the Professor Hélio Fraga National Reference Laboratory. All the DST was performed or reviewed by National Reference Laboratory, which follows the international quality performance standards, proposed by WHO [10]. In 2014, the Xpert MTB / RIF molecular test began to be used in Rio de Janeiro. Until this time, the DST were only indicated for patients with previous treatment, resistant TB case contacts; positive smear at the end of the 2nd month of drug treatment and failure. After 2015, regardless of rifampicin resistance, every case with presumed drug resistance suspected should had culture and DTS performed [11].

## Operational definitions of key terms

**Cured:** The patient should have at least three negative cultures after the 12th month of treatment.

**Treatment completed:** Patient who completed the time stipulated for treatment, with favorable clinical and radiological evolution, but without the cultures of follow up.

**Failure:** Two or more positive cultures out of three recommended after the 12th month of treatment, or three consecutive positive cultures after the 12th month of treatment, at least 30 days apart. It may also be considered according to medical evaluation, and decision to change treatment early due to clinical and radiological worsening.

**Default:** When the treatment was discontinued for 30 consecutive months or more.

**Death:** When the patient died for any reason during the treatment.

**Unfavourable outcome:** The sum of patients who had the outcome classified as death, failure or default.

Treatment success: The sum of patients who had the outcome classified as cured and completed treatment.

## Statistical methods

Given the nature of the dependent variable, number (frequency) and median were used to describe the characteristics of the patients in general and, specifically, of MDR/XDR-TB cases.

Bivariate logistic regressions were performed to evaluate the relationship between each of the treatment outcomes and the following covariates: gender, being less than 40 years, having less than 8 years of schooling, race/color, HIV infection, diabetes, comorbidities (viral hepatitis, renal insufficiency, neoplasia, silicosis, transplanted, mental disorder, prolonged use of corticosteroids, users of TNF alpha inhibitors, seizure, and undefined others), drug use, alcohol dependence, smoking, unemployment, drug resistance category (MDR-TB or XDR-TB), treatment regimen (standardized or individualized), type of resistance (primary, that is, patients with no history of previous treatment; or acquired, meaning patients already treated for TB for 1 month or more) [12], extent of disease (presence of chest cavity and bilateral disease), six-month culture conversion and previous MDR-TB treatment.

The variable six-month culture conversion was created for analysis. It was applied to patients who had at least two negative cultures until the sixth month after the start of treatment.

Patients who had more than one treatment for multidrug resistance registered in SITETB were considered as having previous treatment for MDR-TB.

As the number of results reported for second-line drug susceptibility testing is relatively small, drug resistance was only described. Variables with significance levels ≤0.20 in the univariate analysis were included in the multivariate logistic regression models. Statistical analyses were performed with the STATA program version 13.1.

## Results

### Descriptive analysis

Between 2000 and 2016, 2,477 cases of MDR-TB were reported in RJ, with 208 cases excluded (Fig 1).

Table 1 shows the sociodemographic and clinical variables related to patients with MDR-TB and XDR-TB. Of the total of the 2,269 cases included, 2,129 (93.8%) were MDR-TB and 140 (6.1%) XDR-TB, of which 1466 (64.4%) were men. The overall median age was 38 years (41 years among men and 34 years among women). Regarding race/skin color, 1,372 (60.4%) were Afro-Brazilians and 1,422 (62.6%) had less than 8 years of schooling.

Of the 2,103 patients with known serology for HIV, 167 (7.9%) were positive. There was a higher percentage of HIV infection among TB-XDR patients than among MDR-TB (9.8% and 7.8%, respectively). XDR-TB patients had a higher percentage of comorbidities recorded (excluding diabetes mellitus and HIV) than patients with MDR-TB (14.1% and 11.5%). Among the 247 patients with comorbidities, the most prevalent were undefined others (79,3%), mental disorder (14,2%), neoplasia (8,1%), and viral hepatitis (6,5%). Unemployment was the most frequent risk factor in the two groups, with 344 cases (16.2%) in MDR-TB

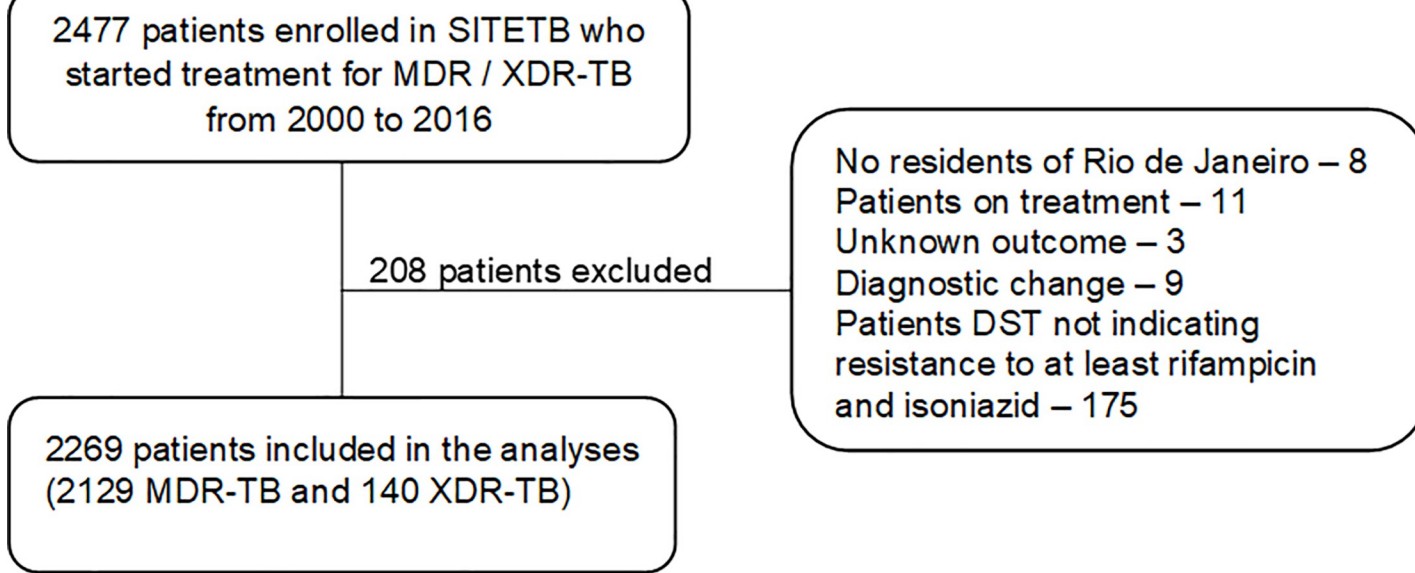

**Fig 1. Flowchart of the study's population.** MDR-TB = multidrug-resistant TB; XDR-TB = extensively drug-resistant TB; SITETB = Special Tuberculosis Treatment System; DST = drug-susceptibility testing.

patients and 29 (20.7%) in XDR-TB ones. Primary drug resistance in patients with MDR-TB and XDR-TB was, respectively, 324 (15.3%) and 10 (7.1%) cases.

Only cavity or bilateral chest X-ray, type of resistance and previous treatment for multidrug resistance showed statistically significant differences between groups of patients.

### Drug-susceptibility and treatment outcomes

Overall, 960 patients, in addition to rifampicin and isoniazid, underwent DST for all first-line drugs and 502 for, at least, one fluoroquinolone and one injectable drug. Among patients who had positive results from the DST for first-line drugs, 703 (69.7%) were resistant to pyrazinamide, 862 (39.8%) to ethambutol and 949 (43.7%) to streptomycin. Among the second-line aminoglycosides, resistance to amikacin was 24.8% (81.8% among XDR patients). Ofloxacin was the most tested fluoroquinolone, with 100% resistance found among patients with XDR-TB (Table 2).

Overall, 1,264 (55.7%) patients had outcome for cure or complete treatment. Unfavourable outcome was observed in 1,005 (44.3%) cases: 433 (19.1%) defaulted, 347 (15.3%) died and 225 (9.9%) failed. Among the most frequent reasons for unfavourable treatment, default occurred in 414 (19.4%) patients with MDR-TB and failure in 53 (37.7%) XDR-TB patients. (Table 3).

### Bivariate-model factors associated with treatment outcomes

The unfavourable outcome was more likely in patients under 40 years of age, who had less than 8 years of schooling, who were Afro-Brazilians, HIV-positive, drug users, unemployed, who did not receive standardized regimens, who had acquired resistance, bilateral and cavitary disease, and who had previous multidrug resistant treatment and TB-XDR. Default was associated with being male, under the age of 40, less than 8 years of schooling, Afro-Brazilians, HIV positive, drug users, alcoholism, smoking, unemployment, acquired resistance, with cavity and previous treatment for multidrug resistance. Having comorbidities and diabetes were protective factors for unfavourable outcome and failure. Death was associated with less than 8 years of schooling, HIV positive, presence of comorbidities, acquired resistance, previous treatment for multidrug resistance, bilateral disease and TB-XDR. Six-month culture conversion was a protective factor for all outcomes (Table 4).

### Multivariate-model factors associated with treatment outcomes

In the final model, being less than 40 years, having less than 8 years of schooling, being an Afro-Brazilian, being a drug user were found to have an association with unfavourable outcome and default. Being HIV positive was associated with unfavourable outcome and death. Bilateral disease and previous treatment for MDR-TB had nearly twice as likely to result in an unfavourable outcome, while XDR-TB had a 4.7-fold higher odds than MDR-TB. Default were associated with being male, smoking and previous treatment for MDR-TB. Drug use was twice as likely to result in default. Bilateral disease, comorbidities and XDR-TB were associated with twice as many chances for death. Six-month culture conversion was found to be a protective factor for all outcomes, but mainly with the unfavourable outcome and death (Table 5).

## Discussion

The study of 2,269 cases found an overall therapeutic success rate of 55.7%, 58.1% among MDR-TB cases and 18.6% among XDR-TB. In 2015, at the global level, the success rate was 55% for MDR-TB cases and 35% for XDR-TB [1].

**Table 1. Demographic and clinical characteristics of 2269 patients with MDR-TB and XDR-TB.**

| Characteristics | Total of patients (%) N = 2269 | MDR-TB (%) N = 2129 | XDR-TB (%) N = 140 | p-value [*] |
|---|---|---|---|---|
| **Sex** | | | | |
| Female | 803 (35.4) | 758 (35.6) | 45 (35.4) | 0.407 |
| Male | 1466 (64.4) | 1371 (64.4) | 95 (67.9) | |
| **Age Range** | | | | |
| 0–24 | 345 (15.2) | 329 (15.5) | 16 (11.4) | 0.310 |
| 25–44 | 1126 (49.6) | 1047 (49.2) | 79 (56.4) | |
| 45–64 | 715 (31.5) | 676 (31.8) | 39 (27.9) | |
| ≥65 | 83 (3.7) | 77 (3.6) | 6 (4.3) | |
| **Ethnical group** | | | | |
| Caucasian | 816 (36.0) | 762 (35.8) | 54 (38.6) | 0.484 |
| Afro-Brazilian | 1372 (60.5) | 1290 (60.6) | 82 (58.6) | |
| Unknown | 81 (3.5) | 77 (3.6) | 4 (2.8) | |
| **Years of study** | | | | |
| None | 113 (5.0) | 109 (5.1) | 4 (2.9) | 0.221 |
| 1 a 3 | 430 (19.0) | 405 (19.0) | 25 (17.9) | |
| 4 a 7 | 879 (38.7) | 828 (38.9) | 51 (36.4) | |
| 8 a 11 | 491 (21.6) | 450 (21.1) | 41 (29.3) | |
| ≥12 | 166 (7.3) | 155 (7.3) | 11 (7.9) | |
| Unknown | 190 (8.4) | 182 (8.6) | 8 (5.7) | |
| **Site of disease** | | | | |
| Extrapulmonary | 20 (0.9) | 20 (0.9) | 0 | 0.403 |
| Pulmonary | 2218 (97.7) | 2079 (97.7) | 139 (99.3) | |
| Both | 31 (1.4) | 30 (1.4) | 1 (0.8) | |
| **HIV status (n = 2103)** | | | | |
| Negative | 1936 (92.1) | 1826 (92.2) | 110 (90.1) | 0.425 |
| Positive | 167 (7.9) | 155 (7.8) | 12 (9.8) | |
| **Risk factors** | | | | |
| Diabetes | 219 (9.6) | 200 (9.4) | 19 (13.5) | 0.105 |
| Alcohol abuse | 269 (11.9) | 257 (12.1) | 12 (8.6) | 0.215 |
| Drug use | 179 (7.9) | 167 (7.8) | 12 (8.6) | 0.757 |
| Smoking | 188 (8.3) | 175 (8.2) | 13 (9.3) | 0.658 |
| Prisoner | 36 (1.6) | 36 (1.7) | 0 | 0.121 |
| Unemployed | 373 (16.4) | 344 (16.2) | 29 (20.7) | 0.159 |
| Comorbidities[†] | 247 (11.7) | 229 (11.6) | 18 (14.1) | 0.391 |
| **Chest radiography (n = 2191)** | | | | |
| Cavitation | 1772 (80.9) | 1652 (80.4) | 120 (88.2) | 0.024 |
| Bilateral | 1648 (75.3) | 1531 (74.6) | 117 (86.0) | 0.003 |
| **Resistance type** | | | | |
| Primary | 334 (14.7) | 324 (15.2) | 10 (7.1) | 0.009 |
| Acquired | 1935 (85.3) | 1805 (84.8) | 130 (92.9) | |
| **Previous MDR-TB treatments** | | | | |
| No | 1744 (76.9) | 1703 (80.0) | 41 (29.3) | <0.001 |
| Yes | 525 (23.1) | 426 (20.0) | 99 (70.7) | |

TB = tuberculosis; MDR-TB = multidrug-resistant TB; XDR-TB = extensively drug-resistant TB

HIV = human immunodeficiency virus.

[*] Comparison between MDR/XDR-TB.

[†] Except Diabetes and HIV.

**Table 2. Drug resistance among 2269 patients with MDR-TB and XDR-TB.**

| Drug | Resistant | MDR-TB | XDR-TB | p-value [†] |
|---|---|---|---|---|
| | n/N* (%) | n/N* (%) | n/N* (%) | |
| **First-line drugs** | | | | |
| Isoniazid | 2269/2269 (100.0) | | | |
| Rifampicin | 2269/2269 (100.0) | | | |
| Ethambutol | 862/2169 (39.7) | 767/2031 (37.8) | 95/138(68,8) | <0.001 |
| Pyrazinamide | 703/1008 (69.7) | 659/958 (68.8) | 44/50 (88) | 0.004 |
| Streptomycin | 949/2160 (43.9) | 858/2022 (42.4) | 91/138 (65.9) | <0.001 |
| **Second-line drugs** | | | | |
| Ethionamide | 303/705 (43.0) | 290/679 (41.7) | 13/26 (50.0) | 0.461 |
| Amikacin | 125/503 (24.8) | 13/366 (3.6) | 112/137 (81.8) | <0.001 |
| Kanamycin | 82/383 (21.4) | 4/280 (1.4) | 78/103 (75.7) | <0.001 |
| Capreomycin | 82/382 (21.5) | 4/280 (1.4) | 78/102 (76.5) | <0.001 |
| Ofloxacin | 372/512 (46.9) | 99/372 (26.6) | 140/140 (100.0) | <0.001 |
| Ciprofloxacin | 5/10 (50.0) | 5/9 (55.6) | 0/1 (0.0) | 0.292 |
| Moxifloxacin | 13/20 (0.0) | 8/14 (57.1) | 5/6 (83.3) | 0.260 |
| Levofloxacin | 4/9 (44.4) | 2/6 (33.3) | 2/3 (66.7) | 0.343 |

MDR-TB = multidrug-resistant TB.

n/N* = number of resistance patients/total of tested patients.

[†]Comparison between MDR/XDR-TB.

Recently, in a meta-analysis of 74 studies including 17,494 participants, treatment success was 26% in patients with XDR-TB and 60% in patients with MDR-TB [13]. Another study conducted in Brazil reports a success rate of 60%, with lower chances of therapeutic success for patients in the Southeast and Northeast regions [7].

Socioeconomic factors are closely linked to the health-disease process and are important predictors of the outcome of tuberculosis. Having less than 8 years of schooling, being less than 40 years, and being Afro-Brazilian and male are associated with the unfavourable outcome and default. This pattern is similar to that observed in the countries of the European Union [14]. In addition, the predominant age group is young people, who are often excluded from the labor market at their most productive age because of the sequelae of the disease [15]. By its turn, poor schooling may restrict understanding of the disease, leading to errors in treatment. Such errors are related mostly to the inappropriate use of medications, which makes the compliance with routines and performance tests more difficult, thus contributing to unfavourable outcomes [16].

**Table 3. Treatment outcomes among 2269 patients with MDR-TB and XDR-TB.**

| Categories of drug resistance | Outcomes | | | | | | |
|---|---|---|---|---|---|---|---|
| | Cured | Treatment completed | Died | Failed | Defaulted | Total | p-value |
| **MDR-TB** | 607 (28.6) | 631 (29.6) | 305 (14.3) | 172 (8.1) | 414 (19.5) | 2129 (93.8) | <0.001* |
| **XDR-TB** | 15 (10.7) | 11 (7.9) | 42 (30.0) | 53 (37.9) | 19 (13.6) | 140 (6.2) | |
| Total | 622 (27.4) | 642 (28.3) | 347 (15.3) | 225 (9.9) | 433 (19.1) | 2269 (100) | |

MDR-TB = multidrug-resistant TB; XDR-TB = extensively drug-resistant TB.

* Comparison of treatment completed and unfavourable outcome by Chi-square test between MDR-TB and XDR-TB.

**Table 4. Univariate analysis: Predictors of unfavourable outcome, default and death among 2269 patients with MDR-TB and XDR-TB.**

| Predictors | n | Unfavourable outcome | | Default | | Death | |
|---|---|---|---|---|---|---|---|
| | | OR (95% CI) | *p*-value | OR (95% CI) | *p*-value | OR (95% CI) | *p*-value |
| **Sex** | | | | | | | |
| Female | 803 | 1.0 | | 1.0 | | 1.0 | |
| Male | 1466 | 1.11 (0.93–1.23) | 0.237 | 1.29 (1.03–1.62) | 0.0024 | 0.95 (0.75–1.21) | 0.697 |
| **≥40 years** | | | | | | | |
| Yes | 1238 | 1.0 | | 1.0 | | 1.0 | |
| No | 1031 | 1.28 (1.08–1.52) | 0.004 | 1.56 (1.26–1.95) | <0.001 | 0.95 (0.75–1.19) | 0.678 |
| **Years of study** | | | | | | | |
| ≥ 8 years | 657 | 1.0 | | 1.0 | | 1.0 | |
| < 8 years | 1422 | 1.77 (1.46–2.14) | 0.001 | 1.84 (1.42–2.39) | <0.001 | 1.67 (1.25–2.24) | <0.001 |
| **Afro-Brazilian** | | | | | | | |
| No | 816 | 1.0 | 0.001 | 1.0 | | 1.0 | |
| Yes | 1372 | 1.40 (1.17–1.67) | | 1.65 (1.30–2.08) | <0.001 | 1.01 (0.79–1.29) | 0.890 |
| **HIV status** | | | | | | | |
| Negative | 1936 | 1.0 | | 1.0 | | 1.0 | |
| Positive | 167 | 1.43 (1.04–1.97) | 0.024 | 1.46 (1.01–2.12) | 0.044 | 1.48 (0.99–2.19) | 0.050 |
| **Diabetes** | | | | | | | |
| No | 2050 | 1.0 | | 1.0 | | 1.0 | |
| Yes | 219 | 0.72 (0.53–0.98) | 0.036 | 0.39 (0.24–0.63) | <0.001 | 0.72 (0.47–1.11) | 0.140 |
| **Comorbidities*** | | | | | | | |
| No | 1864 | 1.0 | | 1.0 | | 1.0 | |
| Yes | 247 | 0.71 (0.53–0.95) | 0.022 | 0.48 (0.31–0.73) | 0.001 | 1.49 (1.07–2.08) | 0.017 |
| **Drug use** | | | | | | | |
| No | 2090 | 1.0 | | 1.0 | | 1.0 | |
| Yes | 179 | 2.97 (1.44–2.69) | <0.001 | 3.13 (2.27–4.32) | <0.001 | 0.76 (0.47–1.20) | 0.246 |
| **Alcohol abuse** | | | | | | | |
| No | 2000 | 1.0 | | 1.0 | | 1.0 | |
| Yes | 269 | 1.18 (0.91–1.52) | 0.198 | 1.55 (1.15–2.08) | 0.004 | 1.03 (0.72–1.46) | 0.876 |
| **Smoking** | | | | | | | |
| No | 2081 | 1.0 | | 1.0 | | 1.0 | |
| Yes | 188 | 1.14 (0.84–1.54) | 0.380 | 1.75 (1.25–2.46) | 0.001 | 0.49 (0.29–0.83) | 0.008 |
| **Unemployed** | | | | | | | |
| No | 1896 | 1.0 | | 1.0 | | 1.0 | |
| Yes | 373 | 1.60 (1.28–2.00) | <0.001 | 1.54 (1.19–2.01) | 0.001 | 1.10 (0.81–1.48) | 0.534 |
| **Categories of drug resistance** | | | | | | | |
| MDR-TB | 2129 | 1.0 | | 1.0 | | 1.0 | |
| XDR-TB | 140 | 6.09 (3.94–9.41) | <0.001 | 0.65 (0.39–1.06) | 0.089 | 2.56 (1.75–3.75) | <0.001 |
| **Six-month culture conversion** | | | | | | | |
| No | 1451 | 1.0 | | 1.0 | | 1.0 | |
| Yes | 818 | 0.18 (0.15–0.23) | <0.001 | 0.42 (0.33–0.54) | <0.001 | 0.10 (0.06–0.16) | <0.001 |
| **Treatment regimen** | | | | | | | |
| Standardized | 1497 | 1.0 | | 1.0 | | 1.0 | |
| Individualized | 766 | 1.56 (1.31–1.86) | <0.001 | 1.02 (0.82–1.28) | 0.811 | 1.02 (0.82–1.28) | 0.811 |
| **Resistance type** | | | | | | | |
| Primary | 334 | 1.0 | | 1.0 | | 1.0 | |
| Acquired | 1935 | 2.02 (1.57–2.59) | <0.001 | 1.81 (1.28–2.56) | 0.001 | 1.58 (1.09–2.28) | 0.014 |
| **Chest radiography** | | | | | | | |

(*Continued*)

**Table 4.** (Continued)

| Predictors | n | Unfavourable outcome | | Default | | Death | |
|---|---|---|---|---|---|---|---|
| | | OR (95% CI) | *p*-value | OR (95% CI) | *p*-value | OR (95% CI) | *p*-value |
| **Sex** | | | | | | | |
| No cavitation | 419 | 1.0 | | 1.0 | | 1.0 | |
| Cavitation | 1772 | 1.62 (1.30–2.02) | <0.001 | 1.45 (1.08–1.96) | 0.012 | 1.01 (0.75–1.35) | 0.952 |
| Unilateral | 541 | 1.0 | | 1.0 | | 1.0 | |
| Bilateral | 1648 | 1.97 (1.60–2.42) | <0.001 | 1.07 (0.81–1.37) | 0.585 | 2.49 (1.78–3.47) | <0.001 |
| **Previous MDR-TB treatment** | | | | | | | |
| No | 1744 | 1.0 | | 1.0 | | 1.0 | |
| Yes | 525 | 3.18 (2.59–3.90) | <0.001 | 1.90 (1.51–2.39) | <0.001 | 1.45 (1.12–1.87) | 0.004 |

TB = tuberculosis; MDR-TB = multidrug-resistant TB; XDR-TB = extensively drug-resistant TB

HIV = human immunodeficiency virus.

*Except Diabetes and HIV.

In Brazil, TB treatment is free and only offered in the public health system. All cases are notified and the supply and distribution of first- and second-line medicines are guaranteed by the Ministry of Health [17,18]. While treatment is free, indirect costs generated by, for example, transportation, food and access to services compromise a significant percentage of the income of poorer patients with MDR-TB [19]. Over the past years, our country has expended measures for social protection. While there are relevant conditional cash-transfer social protection policies, and some cities provide vouchers to pay for patients' transport- related expenses, they are not enough to meet patients' needs during treatment. A study on MDR-TB patients in a reference center in RJ showed that only 38% of participants reported being beneficiaries of social protection because of drug-resistant TB. This demonstrates that there are many barriers to obtaining benefits, such as, for example, the demand of prior contribution to the pension system. This demand is not met because many workers do not have a formal work contract. The adoption of social protection measures was associated with a lower risk of incurring total costs of 20% of family income and of impoverishment [20]. Due to the long treatment, affected households are especially vulnerable to the costs related to TB [20]. In a recent meta-analysis carried out in low- and middle-income countries encompassing 25 studies, social protection measures are found to be associated with successful treatment and reduction of default, in addition to being associated with a lower risk of impoverishment [21]. It is thus necessary to ensure that TB patients and affected families receive appropriate replacement of income and other social protection measures.

Regarding TB/HIV co-infection, several factors are associated with a worse treatment outcome, including a worse absorption of TB drugs, which may contribute to the failure of the basic regimen and acquisition of resistance [22]. HIV infection was found to be associated with an unfavourable outcome and death. The values observed in our study are higher than 1.41 for unsuccessful treatment and 1.66 for death, as described in the meta-analysis that evaluated the outcomes of treatment of MDR-TB and XDR-TB among people living with and without HIV [23]. The same study also showed that unsuccessful treatment among people living with HIV is higher in low-income regions (RR 2.23, 95% CI 1, 60–3,11) than in high-income ones (RR 1.22, 95% CI 0.97–1.53). This indicates that socio-environmental issues and lack of access to health care may increase the vulnerability of this group [23].

Among the risk factors studied, the use of illicit drugs and smoking were found to be associated with default. However, it is possible that smoking is a marker for other social and

**Table 5. Multivariate analysis: Predictors of unfavourable outcome, default and death among 2269 patients with MDR-TB and XDR-TB.**

| Predictors | Unfavourable outcome | | Default | | Death | |
|---|---|---|---|---|---|---|
| | OR* (95% CI) | *p*-value | OR* (95% CI) | *p*-value | OR* (95% CI) | *p*-value |
| **Sex** | | | | | | |
| Female | | | 1.0 | | | |
| Male | | | 1.42 (1.08–1.87) | 0.012 | | |
| **≥40 years** | | | | | | |
| Yes | 1.0 | | 1.0 | | | |
| No | 1.32 (1.06–1.66) | 0.013 | 1.74 (1.33–2.26) | <0.001 | | |
| **Years of study** | | | | | | |
| ≥ 8 years | 1.0 | | 1.0 | | | |
| < 8 years | 1.61 (1.25–2.06) | <0.001 | 1.51 (1.12–2.02) | 0.006 | | |
| **Afro-Brazilian** | | | | | | |
| No | 1.0 | | 1.0 | | | |
| Yes | 1.33 (1.05–1.67) | 0.014 | 1.46 (1.11–1.92) | 0.006 | | |
| **Drug use** | | | | | | |
| No | 1.0 | | 1.0 | | | |
| Yes | 1.78 (1.15–2.75) | 0.009 | 2.17 (1.42–3.31) | <0.001 | | |
| **Smoking** | | | | | | |
| No | | | 1.0 | | | |
| Yes | | | 1.66 (1.06–2.61) | 0.026 | | |
| **Categories of drug resistance** | | | | | | |
| MDR-TB | 1.0 | | 1.0 | | 1.0 | |
| XDR-TB | 4.71 (2.67–8.33) | <0.001 | 0.42 (0.23–0.78) | 0.006 | 2.54 (1.36–3.01) | <0.001 |
| **Resistance type** | | | | | | |
| Primary | | | | | | |
| Acquired | | | | | | |
| **Chest radiography** | | | | | | |
| Unilateral | 1.0 | | | | 1.0 | |
| Bilateral | 2.2 (1.70–2.91) | <0.001 | | | 2.23 (1.50–3.30) | <0.001 |
| **HIV status** | | | | | | |
| Negative | 1.0 | | | | 1.0 | |
| Positive | 1.60 (1.05–2.43) | 0.026 | | | 1.74 (1.10–2.74) | 0.017 |
| **Comorbidities** | | | | | | |
| No | | | 1.0 | | 1.0 | |
| Yes | | | 0.39 (0.22–0.67) | 0.001 | 2.03 (1.36–3.01) | <0.001 |
| **Six-month culture conversion** | | | | | | |
| No | 1.0 | | 1.0 | | 1.0 | |
| Yes | 0.17 (0.13–0.22) | <0.001 | 0.45 (0.34–0.61) | <0.001 | 0.07 (0.04–0.13) | <0.001 |
| **Previous MDR-TB treatment** | | | | | | |
| No | 1.0 | | 1.0 | | | |
| Yes | 2.35 (1.79–3.09) | <0.001 | 1.91 (1.44–2.53) | <0.001 | | |

MDR-TB = multidrug-resistant TB, XDR-TB = extensively drug resistant TB

* Adjusted odds ratio.

behavioral factors that render default more likely [24]. Drug use also makes adherence to treatment difficult. Even when problems of access to the health system are overcome, adherence to long-term treatment regimens may be particularly problematic for drug users. A research

conducted in Brazil corroborated WHO data showing that approximately 10% of the population of large urban centers consume psychoactive substances. This casts in relief the need for governments to devise targeted treatment strategies for drug users [25].

In addition to the above-mentioned risk factors, having comorbidities was a predictive factor for death and protective for default. One plausible hypothesis is that these patients find in the tertiary care a better resolution for their comorbidities and, therefore, adhere to the treatment at a higher rate. Among the comorbidities reported, the most prevalent was undefined others. Thus, it is key to improve the reporting of specific comorbidities, so as to clarify the mechanisms underlying unfavourable treatment results for different subpopulations. This will allow the development of new strategies for care and more individualized and adequate support for these cases.

Although XDR-TB patients are less likely to discontinue treatment, this paper's data indicates that XDR-TB patients were 4.7 times more likely to have an unfavourable treatment outcome than MDR-TB patients. When resistance was analyzed, all XDR-TB patients were found to be resistant to ofloxacin. Interestingly, a study in Pakistan reported that patients resistant to ofloxacin were 3.2 times more likely to be unsuccessful in treatment [26]. While data on the level of resistance to second-line drugs is still limited, this study shows that increase in resistance to fluoroquinolones can be attributed to its abusive consumption, especially in pneumonia and uncomplicated respiratory tract infections [27,28].

Moreover, among the 140 XDR-TB patients in this study's sample, 41 (29.3%) did not received previous multidrug resistant treatment. That is, they had never been treated with second-line drugs. Despite this, they had strains resistant to drugs, such as capreomycin, amikacin and kanamycin, that are not routinely used for any other disease. In Brazil the treatment of multidrug resistance is carried out only in reference centers and linked to the notification and follow-up of the case in SITE-TB. This means that there is a control of the medication dispensed per patient. Therefore, a plausible hypothesis to explain that 29.3% patients were on the first multidrug resistant treatment is that these patients had been infected with these strains (primary TB-XDR). Notably, in a meta-analysis performed with patients with fluoroquinolone-resistant strains, second-line injectables, or both, only 25% had previously been treated with second-line drugs. The others had been treated with first-line drugs or had never been treated [29].

Although the main hypothesis for increasing resistance cases is related to acquired resistance, which generated by non-adherence to treatment [30], the data evaluated in this study suggests that a high proportion of cases of XDR-TB is due to primary infection in RJ.

Additionally, this study showed that the six-month culture conversion was a protective factor for the three outcomes, especially for the unfavourable and for death. Interestingly, studies conducted with MDR-TB patients found that the conversion status at 6 months was significantly associated with treatment success as compared to failure or death. Among patients with successful treatment, the median time to culture conversion was significantly lower than among those who had unfavourable results [31,32].

Lastly, this study has some limitations. In the SITETB database, up to 2015 the variables related to diabetes, comorbidities, drug use, alcohol dependence and smoking when classified as "no" may also mean lack of information. There is also no standardization for classification of alcohol dependence, smoking and mental health disorders. Another problem in the data is the low number of DST results reported for first- and second-line drugs. Some hypotheses for this are that the Laboratory had not received necessary supplies at certain times or had laboratory technicians to perform and provide DST results; health professionals did not request DTS in retreatment patients, despite the Ministry of Health guidelines; or DTS results were not recorded in the system. This made it impossible to use drug resistance as an independent

variable. In any case, the statistical findings reported here can be said to be robust because of the large sample size on which they are based.

## Conclusion

Unfavourable outcomes in Rio de Janeiro State in 2000–2016 were associated with socioeconomic factors, comorbidities, severity, and extent of the disease. XDR-TB was strongly associated with the unfavourable treatment outcome. The high rates of failure (37.9%) and death (30.0%) in this category reflect the limitations of treatment options and the urgency of Brazil's health system to incorporate new drugs in the treatment of multidrug resistance. This study also points out that ~30% of XDR-TB cases may have occurred through primary transmission. In view of this, efforts should be directed towards increasing cure and reducing default, improving the access of vulnerable populations to health services, expanding social protection measures, and implementing public policies that prevent the emergence of new cases of drug resistance.

## Supporting information

**S1 Dataset. Dataset of research.**
(XLSX)

## Acknowledgments

The authors would like to thank the managers of the Rio de Janeiro State tuberculosis program involved in TB surveillance for their permanent effort to follow up TB cases.

## Author Contributions

**Conceptualization:** Marcela Bhering, Raquel Duarte, Afrânio Kritski.

**Data curation:** Marcela Bhering.

**Formal analysis:** Marcela Bhering.

**Funding acquisition:** Afrânio Kritski.

**Methodology:** Marcela Bhering, Afrânio Kritski.

**Supervision:** Raquel Duarte, Afrânio Kritski.

**Validation:** Afrânio Kritski.

**Visualization:** Marcela Bhering.

**Writing – original draft:** Marcela Bhering.

**Writing – review & editing:** Marcela Bhering, Raquel Duarte, Afrânio Kritski.

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
