## [Decision Letter · Decision Letter 0]

6 Sep 2019

PONE-D-19-15007

Predictive factors for unfavourable treatment in MDR-TB and XDR-TB patients in Rio de Janeiro State, Brazil, 2000-2016

PLOS ONE

Dear Msc Bhering,

Thank you for submitting your manuscript to PLOS ONE. After careful consideration, we feel that it has merit but does not fully meet PLOS ONE’s publication criteria as it currently stands. Therefore, we invite you to submit a revised version of the manuscript that addresses the points raised during the review process.

We would appreciate receiving your revised manuscript by Oct 21 2019 11:59PM. To enhance the reproducibility of your results, we recommend that if applicable you deposit your laboratory protocols in protocols.io, where a protocol can be assigned its own identifier (DOI) such that it can be cited independently in the future. For instructions see: http://journals.plos.org/plosone/s/submission-guidelines#loc-laboratory-protocols

We look forward to receiving your revised manuscript.

Kind regards,

Alejandro Escobar-Gutiérrez, Ph.D.

Academic Editor

PLOS ONE

Journal Requirements:

Additional Editor Comments (if provided):

Please analyze very carefully and respond to all the comments of the reviewer. I agree with such comments and a further consideraton in regard of the importance of a study like this could be of special interest for the PLOS-ONE readers. The success of TB control programmes, specially in regard of the control of MDR and XDR cases, are absolutely necessary to achieve the goal of an effective TB world control. The addition of more detailed description of the procedures, economic support, patients acceptance and results evaluation of your experience in the largest city of Brazil, will improve the quality of your manuscript. This comprehensive information could be a model to improve the TB control programmes implemented in any community where TB still remains as a major public health problem.

Reviewers' comments:

Reviewer's Responses to Questions

**Comments to the Author**

1. Is the manuscript technically sound, and do the data support the conclusions?

Reviewer #1: Yes

2. Has the statistical analysis been performed appropriately and rigorously? 

Reviewer #1: Yes

3. Have the authors made all data underlying the findings in their manuscript fully available?

Reviewer #1: Yes

4. Is the manuscript presented in an intelligible fashion and written in standard English?

Reviewer #1: Yes

5. Review Comments to the Author

Reviewer #1: The manuscript is of interest to the readership of the journal.

I have several concerns. The first is that the manuscript does not describe the characteristics of the tuberculosis control program relevant to this study. Please describe its main aspects. The authors mention that treatment if free of charge. Who treat these patients? Is there a referral and counter-referral system? Who pays for second-line drugs? Is the supply of drugs continuous and sufficient? What are the obstacles that patients may face to access treatment?

My second concern refers to DST procedures. Please describe the laboratory (or laboratories) that performed the DST tests. What determines that a sample is processed (or not) for DST? Are there regional laboratories? What are the quality performance standards? The study spans 16 years during which significant changes might have taken place. It would be useful to describe modifications during the study period.

Finally it would be useful to expand on treatment regimes. The authors describe that treatment schemes were either standardized or individualized. It would be useful to expand on this. Were there changes during the study period? If there were changes during the 16-year duration of the study, it might be useful to consider the different periods to analyze if there were modifications in treatment outcomes.

It would be useful to provide the ID of the IRB that reviewed the manuscript.

Given the scope of PLOS ONE would be useful to provide a brief description of MDR, XDR, and treatment outcomes.

It would be useful to add to the flowchart the number of patients who underwent DST tests and the main reasons that explain why the program did not process all samples.

It would be useful to provide the absolute numbers in Table 4.

6. PLOS authors have the option to publish the peer review history of their article (what does this mean?). If published, this will include your full peer review and any attached files.

Reviewer #1: No

---

## [Author Response · Author response to Decision Letter 0]

16 Oct 2019

I have improved the definition of the key concepts and tried to describe how the tuberculosis control program in Brazil works, as well as clarify the main difficulties for patients.

Reviewer #1: The manuscript is of interest to the readership of the journal.

I have several concerns. 

1) The first is that the manuscript does not describe the characteristics of the tuberculosis control program relevant to this study. Please describe its main aspects. 

2) The authors mention that treatment if free of charge. Who treat these patients? 

3) Is there a referral and counter-referral system? 

To answer questions 1, 2, and 3, I have included the topic “Tuberculosis Control Program, Treatment regimens for MDR-TB and laboratory diagnosis in Brazil” and improved the SITE-TB description (lines 66-72)

4) Who pays for second-line drugs? All cases are notified and the supply and distribution of first- and second-line medicines are guaranteed by the Ministry of Health (lines 271-272)

5) Is the supply of drugs continuous and sufficient? Yes. Several TB medicines are produced in Brazil by Fiocruz. The other drugs, which are imported, already have a budget provided for in the annual budget of the Ministry of Health, based on the projection of the national tuberculosis control program.

6) What are the obstacles that patients may face to access treatment? 

Explained on lines 274-288

“While treatment is free, indirect costs generated by, for example, transportation, food and access to services compromise a significant percentage of the income of poorer patients with MDR-TB [20]. Over the past years, our country has expended measures for social protection. While there are relevant conditional cash-transfer social protection policies, and some cities provide vouchers to pay for patients’ transport- related expenses, they are not enough to meet patients’ needs during treatment. A study on MDR-TB patients in a reference center in RJ showed that only 38% of participants reported being beneficiaries of social protection because of drug-resistant TB. This demonstrates that there are many barriers to obtaining benefits, such as, for example, the demand of prior contribution to the pension system. This demand is not met because many workers do not have a formal work contract. The adoption of social protection measures was associated with a lower risk of incurring total costs of 20% of family income and of impoverishment [21]. Due to the long treatment, affected households are especially vulnerable to the costs related to TB [21].”

My second concern refers to DST procedures. 

7) Please describe the laboratory (or laboratories) that performed the DST tests. 

Lines 104-106

“about laboratory tests, cultures and DST for first line medicines were performed at the Central Laboratory of RJ (LACEN) and DST for second line drugs were performed at the Professor Hélio Fraga National Reference Laboratory.”

8) What determines that a sample is processed (or not) for DST? 

Lines 109-114

“In 2014, the Xpert MTB / RIF molecular test began to be used in Rio de Janeiro. Until this time, the DST were only indicated for patients with previous treatment, resistant TB case contacts; positive smear at the end of the 2nd month of drug treatment and failure. After 2015, regardless of rifampicin resistance, every case with presumed drug resistance suspected should had culture and DTS performed [11].”

9) Are there regional laboratories? 

In Rio de Janeiro state the regional laboratory is called Central Laboratory (LACEN).

10) What are the quality performance standards? 

Lines 107-109

“All the DST was performed or reviewed by National Reference Laboratory, which follows the international quality performance standards, proposed by WHO [10]”

11) The study spans 16 years during which significant changes might have taken place. It would be useful to describe modifications during the study period.

Described in the topic “Tuberculosis Control Program, Treatment regimens for MDR-TB and laboratory diagnosis in Brazil.”

12) Finally, it would be useful to expand on treatment regimes. The authors describe that treatment schemes were either standardized or individualized. It would be useful to expand on this. 

Lines 93-101

“The standardized treatment regimen for MDR-TB is recommended and applied in Brazil and should include four drugs, preferably not previously used, containing: a fluoroquinolone, an injectable drug, two second-line drugs (ethionamide, terizidone, linezolid or clofazimine) and an oral first line drug (ethambutol or pyrazinamide), if susceptible [6].

Individualized regimens are restricted to patients with additional resistance to first-line drugs, pre-XDR, XDR-TB, and to patients who have had adverse events with standardized regimens. These regimens might include other oral drugs, such as clofazimine, linezolid, imipenem and high-dose isoniazid [8].”

13)Were there changes during the study period? 

Described in the topic “Tuberculosis Control Program, Treatment regimens for MDR-TB and laboratory diagnosis in Brazil.”

14) If there were changes during the 16-year duration of the study, it might be useful to consider the different periods to analyze if there were modifications in treatment outcomes.

As the main change described in the treatment during this period was the change of amikacin for streptomycin as the first injectable drug, I think that no modification is necessary since the drugs used in the treatment did not enter the statistical analyses.

15) It would be useful to provide the ID of the IRB that reviewed the manuscript.

I included the topic “Ethics statement” to answer this question (lines 74-81).

16) Given the scope of PLOS ONE would be useful to provide a brief description of MDR, XDR, and treatment outcomes.

I have included the following description in the Introduction: “Resistant multidrug TB (MDR-TB) is defined as TB with resistance to at least rifampin and isoniazid and extensively resistant TB (TB-XDR), such as MDR-TB plus resistance to at least one quinolone, and to one second-line injectable drugs used to treat TB (capreomycin, kanamycin and amikacin).” Lines 45-48

17) It would be useful to add to the flowchart the number of patients who underwent DST tests 

Instead of adding the number of patients to the flowchart, I have included the following information in the topic “Drug-susceptibility and treatment outcomes” (Lines 193-195): “Overall, 960 patients, in addition to rifampicin and isoniazid, underwent DST for all first-line drugs and 502 for, at least, one fluoroquinolone and one injectable drug.”

18) and the main reasons that explain why the program did not process all samples.

Lines 361-366

“Another problem in the data is the low number of DST results reported for first- and second-line drugs. Some hypotheses for this are that the Laboratory had not received necessary supplies at certain times or had laboratory technicians to perform and provide DST results; health professionals did not request DTS in retreatment patients, despite the Ministry of Health guidelines; or DTS results were not recorded in the system.”

19) It would be useful to provide the absolute numbers in Table 4. Done.

---

## [Editor Report · Decision Letter 1]

1 Nov 2019

Predictive factors for unfavourable treatment in MDR-TB and XDR-TB patients in Rio de Janeiro State, Brazil, 2000-2016

PONE-D-19-15007R1

Dear Dr. Bhering,

We are pleased to inform you that your manuscript has been judged scientifically suitable for publication and will be formally accepted for publication once it complies with all outstanding technical requirements.

With kind regards,

Alejandro Escobar-Gutiérrez, Ph.D.

Academic Editor

PLOS ONE
---

## [Editor Report · Acceptance letter]

7 Nov 2019

PONE-D-19-15007R1 

Predictive factors for unfavourable treatment in MDR-TB and XDR-TB patients in Rio de Janeiro State, Brazil, 2000-2016 

Dear Dr. Bhering:

I am pleased to inform you that your manuscript has been deemed suitable for publication in PLOS ONE. Congratulations! Your manuscript is now with our production department. 

With kind regards,

on behalf of

Dr. Alejandro Escobar-Gutiérrez 

Academic Editor

PLOS ONE